# Modulation of DNA Methylation/Demethylation Reactions Induced by Nutraceuticals and Pollutants of Exposome Can Promote a C > T Mutation in the Breast Cancer Predisposing Gene *PALB2*

Florestan Courant [1,2], Gwenola Bougras-Cartron [1,2,3], Caroline Abadie [3], Jean-Sébastien Frenel [1,2,3] and Pierre-François Cartron [1,2,3,*]

1    Nantes Université, Inserm, CNRS, Université d'Angers, CRCI2NA, 44000 Nantes, France
2    SIRIC ILIAD, 44000 Nantes, France
3    Institut de Cancérologie de l'Ouest, 44800 Saint-Herblain, France
*    Correspondence: pierre-francois.cartron@inserm.fr

**Abstract:** Background: Deregulation of DNA methylation/demethylation reactions may be the source of C > T mutation via active deamination of 5-methylcytosine to thymine. Exposome, that is to say, the totality of exposures to which an individual is subjected during their life, can deregulate these reactions. Thus, one may wonder whether the exposome can induce C > T mutations in the breast cancer-predisposing gene *PALB2*. Methods: Our work is based on the exposure of MCF10A mammary epithelial cells to seven compounds of our exposome (folate, Diuron, glyphosate, PFOA, iron, zinc, and ascorbic acid) alone or in cocktail. The qMSRE and RMS techniques were used to study the impact of these exposures on the level of methylation and mutation of the *PALB2* gene. Results: Here, we have found that exposome compounds (nutriments, ions, pollutants) promoting the cytosine methylation and the 5-methylcytosine deamination have the ability to promote a specific C > T mutation in the *PALB2* gene. Interestingly, we also noted that the addition of exposome compounds promoting the TET-mediated conversion of 5-methylcytosine (Ascorbic acid and iron) abrogates the presence of C > T mutation in the *PALB2* gene. Conclusions: Our study provides a proof of concept supporting the idea that exposomes can generate genetic mutation by affecting DNA methylation/demethylation.

**Keywords:** epigenetics; DNA methylation; C > T mutation; exposome; *PALB2*; breast cancer

## 1. Introduction

DNA methylation is an epigenetic process involving the transfer of a methyl group (CH3) to the 5′ position of a cytosine from S-adenyl methionine (SAM) molecules. Proteins of the DNA methyltransferase family or DNMTs are the enzymes catalyzing this transfer [1]. In contrast, DNA demethylation can be defined as the epigenetic process resulting in the loss of the methyl group (CH3) at the 5′ position of a cytosine. This loss can be passive by the absence of maintenance of the DNA methylation profile or active, that is, catalyzed by enzymes such as proteins of the TETs (Ten eleven translocation) family, those of the AID/APOBEC (cytidine deaminase/apolipo-protein B mRNA-editing enzyme) proteins, or by the TDG (Thymine-DNA glycosylase) protein [2,3].

Literature reports several crosstalks between DNA methylation/demethylation and genetic alterations: (i) hypomethylation of retrotransposons that promotes chromosomal instability [4–6], (ii) mutations in genes encoding for enzymes governing the DNA methylation/demethylation reactions (such as the one occurring in DNMT3a or TET2 as an example) [7,8], and (iii) C > T mutations occurring via the deamination of 5-methylcytosine whether this reaction is spontaneous or catalyzed by the APOBEC enzyme, for example [9,10].

As reported by Alexendrov et al., (2020), C > T mutation is part of the repertoire of mutational signatures in human cancer [11]. This type of mutation is classified as one of the six listed single-base substitutions or SBS. In addition, several C > T mutations are defined as driver mutations, that is, as "changes" in the DNA sequence of genes that cause cells to become cancer cells and grow and spread in the body. Characterizing driver mutations in tumor tissue may help plan treatment to stop cancer cells from growing, including drugs that target a specific mutation (NCI's dictionary of cancer terms, https: //www.cancer.gov/publications/dictionaries/cancer-terms (accessed on 16 August 2022)). Today, several molecular mechanisms are proposed to be at the origin of driver C > T mutations including spontaneous and AID/APOBEC-catalyzed deamination of 5 mC and defective DNA mismatch repair [12]. C > T mutations have been detected in a large number of genes including $ERBB2^{c.929C>T}$, $PTEN^{c697C>T}$, $TP53^{c637C>T}$, $ARID1A^{c.5965C>T}$, $IDH1^{c.394C>T}$, $BRCA1^{c.4183C>T}$ and $BRC2A^{c.8827C>T}$. C > T mutations were also detected in the *PALB2* gene with the c.1240C > T [13], c.2257C > T [14], c.3256C > T [15], and c.1027C > T mutations [16]. In addition to being detected in breast cancer at a somatic level, note that the PALB2 gene is considered a high-risk breast cancer gene if mutated at a germline level [17–19].

The exposome, that is, the totality of exposures (chemical, microbiological, physical, and medicinal environment, food, ....) to which an individual is subjected during their life, has the capacity to affect DNA methylation/demethylation reactions, and by ricochet, the presence of C > T mutations, putatively [20]. Indeed, the literature shows that the expression level of several enzymes catalyzing DNA methylation/demethylation reactions can be affected by the chemical environment of an individual: glyphosate (CAS No: 1071-83-6) can induce the overexpression of TET3 [21], PFOA (CAS No: 335-67-1) modulates the expression of epigenetic enzymes such as DNMTs), TETs, TDG, or some HDACs (Histone deacetylase) [22]. Diuron (CAS No:330-54-1) modulates the expression of some APOBEC proteins [23]. Food and nutraceuticals also have the ability to modulate DNA methylation/demethylation reactions by modulating the supply of cofactors and co-substrates of enzymes catalyzing these reactions (as introduced above) [24]. The literature shows that folic acid induces a gain in DNA methylation since folic acid contributes to the supply of methyl groups for DNMTs via S-adenosylmethionine (SAM) production [25–27]. Several ions act as co-factors for DNA methylation/demethylation enzymes. TET enzymes that hydroxylate 5 mC to 5 hmC utilize iron ions as a cofactor [28]. Ascorbic Acid is also described to enhance TET-mediated DNA demethylation [29].

In this work, we provide a proof of concept supporting the idea that different compounds of exposome have the ability to guide the DNA methylation/demethylation reactions to promote C > T mutation in the *PALB2* gene. In our work, the *PALB2* gene choice as a "demonstrator gene" is based on the fact that this gene is methylated or mutated in different types of cancer including breast cancer [17,30–33]. Thus, the results obtained for this "demonstrator gene" may be translated to other genes.

## 2. Results

In this work, we hypothesized that exposome compounds could promote C > T mutation in the *PALB2* gene by enhancing or inhibiting DNA methylation/demethylation reactions (Figure 1).

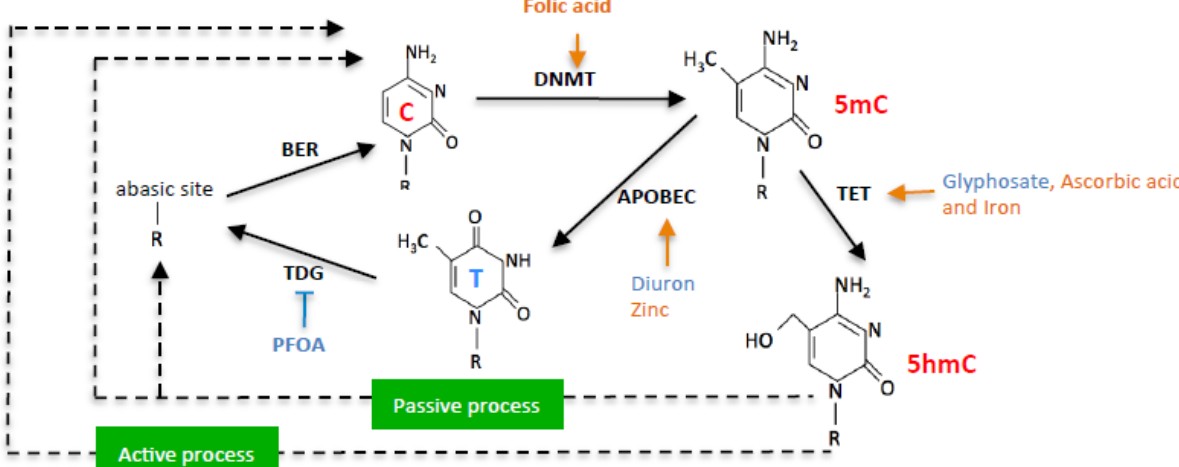

**Figure 1.** Schematic representation of nutraceuticals (orange) and pollutants (blue) effects on DNA methylation/demethylation reactions.

### 2.1. Folic Acid Supplementation Promotes the Methylation of PALB2 Gene Region Susceptible to Promote the c.1027C > T Mutation

Our first hypothesis is based on the fact that Folate could induce the methylation of cytosine giving birth to C > T mutations in the *PALB2* gene. To investigate this hypothesis, we exposed epithelial breast cells (MCF-10A cells) for 3 weeks to three doses of Folate equal to the recommended daily intake (RDI), three times RDI (3RDI), and 10 times RDI (10RDI) [34] (Figure 2). As expected, the quantification of 5-methylcytosine (5 mC) by ELISA method indicated that Folate increased in a dose-dependent manner the global 5 mC level (Figure 3A). Next, we analyzed whether the Folate-induced cytosine methylation could occur on cytosine giving birth to C > T mutations seen in the *PALB2* gene. Based on the ClinVar Miner database, we focused our study on 4 C > T mutations: c.1027C > T, c.1240C > T, c.2257C > T, and c.3256C > T. The choice of these four mutations is based on the fact that these mutations must be included in a "methylable" CG dinucleotide. In practice, the Folate-induced methylation of cytosine was then analyzed using the MeDIP method. The MeDIP assay showed that only the c.1027C > T mutation-prone region was methylated following Folate supplementation at the indicated RDI (Figure 3B). Based on this finding, next, we focused our study on the c.1027C > T mutation in PALB2 gene (*PALB2*$^{c.1027C > T}$).

### 2.2. Diuron and PFOA Supplementation Affect APOBEC Expression and TDG Activity, Respectively

Our second hypothesis is based on the idea of using exposome compounds to guide the DNA demethylation reaction toward 5 mC deamination while blocking the action of the TDG enzyme, that is, the enzyme that initiates T/G mismatch repair (Figure 1). To investigate this hypothesis, we used Diuron and PFOA as two exposome compounds since these both compounds are already known or suspected to modulate DNA methylation levels [23,35,36] (Figure 2). About our working dose, we used concentration ranges based on the consideration of the maximum allowable concentrations (MAC) of these both compounds [37,38]. Thus, our data indicated that Diuron increased expressions of APOBECs enzymes (Figure 4A) and PFOA decreased the TDG activity in a dose-dependent manner (Figure 4B).

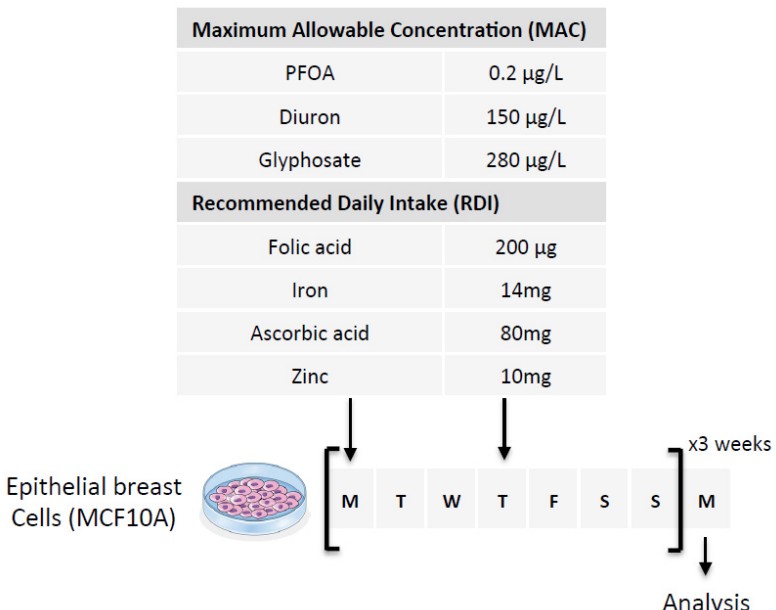

**Figure 2.** Schematic representation of the experimental design used. Nutraceutical and/or pollutant doses are based on the consideration of Maximum Allowable Concentration (MAC) and Recommended Daily Intake (RDI). The cells were exposed to six doses of the compounds over three weeks (two doses per week for 3 weeks). The first two arrows indicate the days of treatment of the cells and the last arrow indicates the day when the cells were used for analysis.

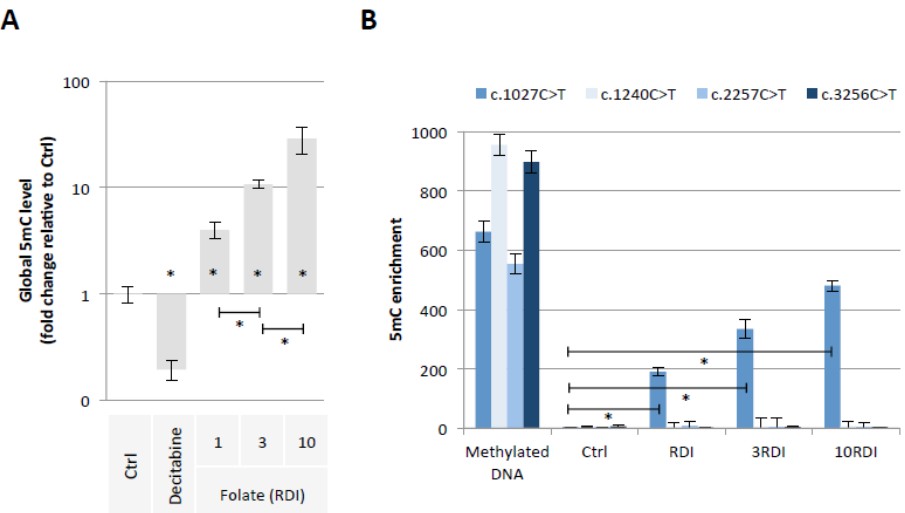

**Figure 3.** Folic acid supplementation at indicated RDI promotes the methylation of *PALB2* gene region susceptible to the c.1027C > T mutation. (**A**) Impact of folic acid on global 5mC level (ELISA method). Graph illustrates the relative fold change of global 5mC level seen in MCF10A cells exposed to Decitabine (DNA demethylating agent used here as an inducer of global DNA hypomethylation, 10 μM) or to folic acid at different doses (RDI and multiple of RDI). "Ctrl" represents the control exposure performed with PBS/DMSO5%. A significant difference (*) was observed between the different doses of folate used. (**B**) Impact of folic acid on methylation level of *PALB2* gene region susceptible to the c.1027C > T mutation (MeDIP method). Graph illustrates the enrichment of 5-methylcytosine (m5C) of *PALB2* gene region susceptible to the c.1027C > T mutation seen in MCF10A cells exposed to folic acid at different doses (RDI and multiple of RDI). "Ctrl" represents the control exposure performed with PBS/DMSO 5%. Methylated DNA: Universal Methylated DNA Standard (Ozyme/Zymo, Saint Cyr L'Ecole, France). A significant difference (*) was observed between control and all investigated conditions.

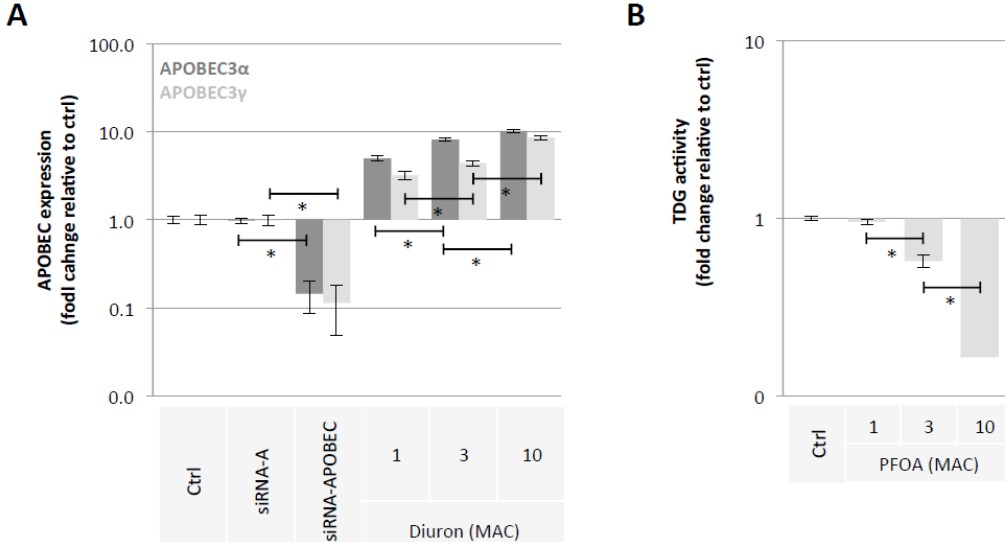

**Figure 4.** Diuron and PFOA supplementation affect APOBEC expression and TDG activity, respectively. (**A**) Impact of Diuron on the APOBEC3α and APOBEC3γ expression (ELISA method). Graph illustrates the fold change expression of APOBEC3α and APOBEC3γ expression seen in MCF10A cells exposed to Diuron at different doses (MAC and multiple of MAC). "Ctrl" represents the control exposure performed with PBS/DMSO 5%. siRNA-A (control) and siRNA-APOBECs are used as control conditions inducing down-expression of APOBEC3α and APOBEC3γ. A significant difference (*) was observed between the different doses of Diuron used. (**B**) Impact of PFOA on TDG activity (Activity assay method). Graph illustrates the fold change of TDG activity seen in MCF10A cells exposed to PFOA at different doses (MAC and multiple of MAC). "Ctrl" represents the control exposure performed with PBS/DMSO5%. A significant difference (*) was observed between the different doses of PFOA used.

### 2.3. A Mixture Composed of Folate, Diuron or Zinc, and PFOA has the Ability to Promote the c.1027C > T Mutation in PALB2 Gene

Based on these findings, we extended our study by using a mixture including Folate, Diuron, and PFOA. A restriction site mutation (RSM) assay was developed to study the presence of $PALB2^{c.1027C > T}$. Our data indicated that the exposure of MCF-10A cells to a mixture including folate at RDI, Diuron at 3 MAC and PFOA at 3 MAC (Folate$^{RDI}$/Diuron$^{3MAC}$/PFOA$^{3MAC}$) promoted $PALB2^{c.1027C > T}$ (Figure 5A).

APOBECs enzymes being zinc (Zn)-dependent deaminases [39], we also analyzed whether the use of Zinc instead of Diuron in Folate$^{RDI}$/Diuron$^{3MAC}$/PFOA$^{3MAC}$ could induce $PALB2^{c.1027C > T}$. Our data indicate that Folate$^{RDI}$/Zn$^{5RDI}$/PFOA$^{3MAC}$ and Folate$^{RDI}$/Zn$^{10RDI}$/PFOA$^{3MAC}$ cocktails promoted $PALB2^{c.1027C > T}$ (Figure 5B).

### 2.4. Ascorbic Acid and Iron Have the Ability to Limit the Presence of the c.1027C > T Mutation in PALB2 Gene

Finally, we hypothesized that some exposome compounds could redirect the DNA demethylation reaction toward TET-mediated DNA demethylation in order to minimize the 5 mC deamination leading to $PALB2^{c.1027C > T}$. For this purpose, we considered glyphosate, ascorbic acid, and iron since these compounds increased the TET3 expression (Figure 6A) or the TET activity (Figure 6B) [40]. Among the tested mixture, we noted that the addition of Ascorbic Acid (AA$^{3RDI}$) and Iron at 3 RDI (Fe$^{3RDI}$) limited the presence of $PALB2^{c.1027C > T}$ induced by Folate$^{RDI}$/Diuron$^{3MAC}$/PFOA$^{3MAC}$ or Folate$^{RDI}$/Zn$^{5RDI}$/PFOA$^{3MAC}$ (Figure 6C). More interestingly, we noted that this effect was not observed in the presence of Folate$^{RDI}$/ Diuron$^{10MAC}$/PFOA$^{10MAC}$ or Folate$^{RDI}$/Zn$^{10RDI}$/PFOA$^{10MAC}$, that is, in the presence of a high concentration of PFOA, Diuron, or Zinc (Figure 6C).

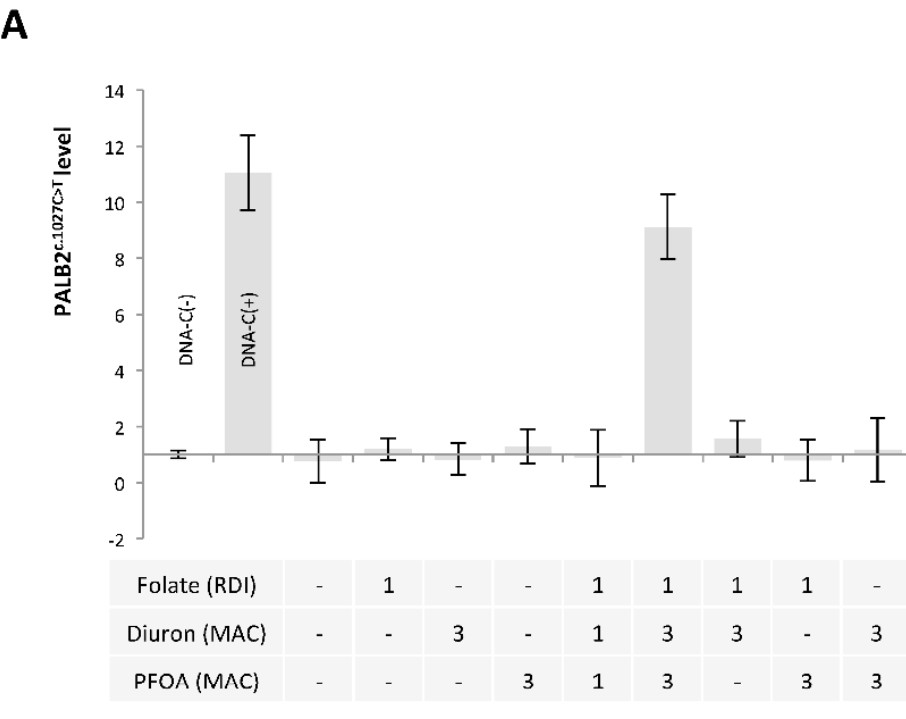

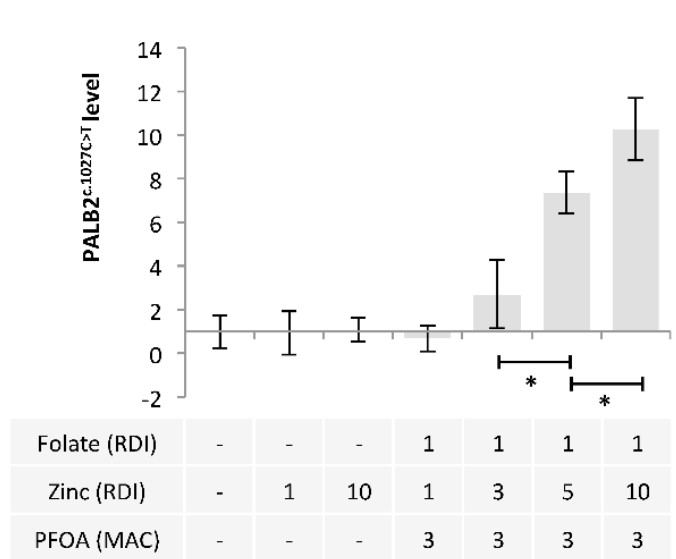

**Figure 5.** A mixture composed of Folate, Diuron or Zinc, and PFOA has the ability to promote the c.1027C > T mutation in *PALB2* gene. (**A,B**) Graphs illustrate the Impact of nutraceuticals and pollutants as single agent or in a mixture on the presence of the c.1027C > T mutation in *PALB2* gene (Restriction site mutation (RSM) method. Doses used are indicated in multiples of RDI or MAC. (*) indicates a significant t test ($p < 0.05$).

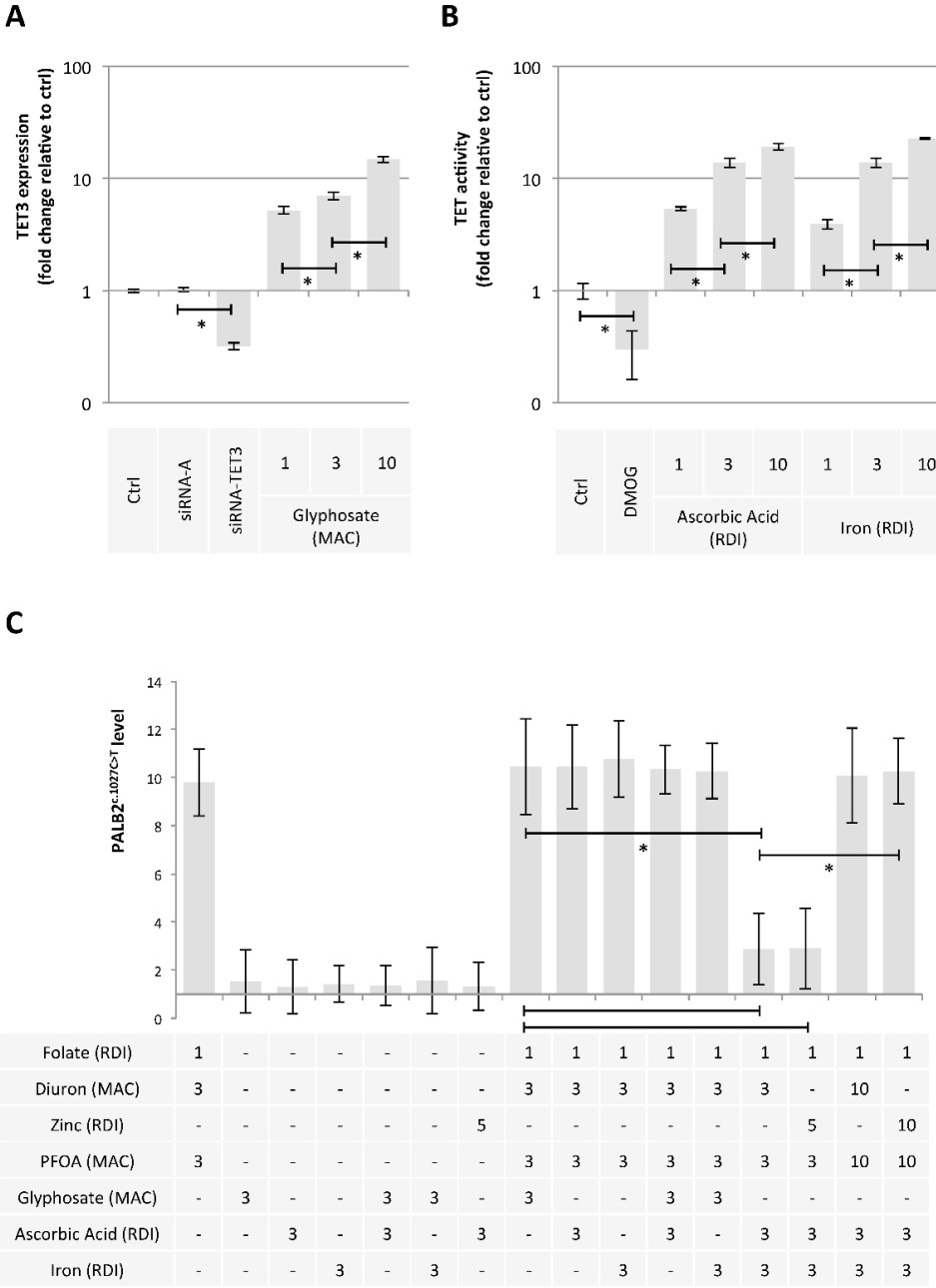

**Figure 6.** Ascorbic acid and Iron have the ability to limit the presence of the c.1027C > T mutation in *PALB2* gene. (**A**) Impact of Glyphosate on TET3 expression (ELISA method). Graph illustrates the fold change expression of TET3 expression seen in MCF10A cells exposed to Glyphosate at different doses (MAC and multiple of MAC). "Ctrl" represents the control exposure performed with PBS/DMSO 5%. siRNA-A (control) and siRNA-TET3 are used as control conditions inducing down-expression of TET3. A significant difference (*) was observed between the different doses of glyphosate used. (**B**) Impact of Ascorbic Acid or Iron on TET activity (Activity assay method). Graph illustrates the fold change of TDG activity seen in MCF10A cells exposed to Ascorbic Acid or Iron at different doses (RDI and multiple of RDI). "Ctrl" represents the control exposure performed with PBS/DMSO 5%. DMOG was used as control condition inducing a decrease in TET activity since DMOG acts as a TET inhibitor. A significant difference (*) was observed between the different doses of ascorbic acid and iron used. (**C**) Graphs illustrate the Impact of nutraceuticals and pollutants as single agent or in a mixture on the presence of the c.1027C > T mutation in *PALB2* gene (Restriction site mutation (RSM) method). Doses used are indicated in multiples of RDI or MAC. A significant difference (*) was observed between the indicated comparisons.

## 3. Discussion

Gene-Environment interaction (GxE) studies are an opportunity to understand the complexity of the interactions between genes and environment, and if necessary to identify predictive biomarkers for the development of pathologies such as cancer. For several years, GxE studies have benefited from the integration of complex data from epidemiological studies and omics-based technology analyses that comprehensively assess an individual's exposome, transcriptome, epigenome, or metabolome [41,42]. Although these approaches allow and will allow the definition of multiple interactions between the exposome and the occurrence of cancers, it remains important to decipher the molecular mechanisms by which the exposome is likely to modify gene integrity and expression. Indeed, a better understanding of these mechanisms could permit the consideration of preventive or medicinal actions counteracting these mechanisms and thus limit the risk of cancer occurrence. It is in this global context that the results of our study are placed while raising different points for discussion.

By reporting that the $Folate^{RDI}/Diuron^{3MAC}/PFOA^{3MAC}$, $Folate^{RDI}/Zn^{5RDI}/PFOA^{3MAC}$, $Folate^{RDI}/Diuron^{3MAC}/PFOA^{3MAC}/Ascorbic$ $Acid^{3RDI}/Iron^{3RDI}$, and $Folate^{RDI}/Zn^{5RDI}/$ $PFOA^{3MAC}/Ascorbic$ $Acid^{3RDI}/Iron^{3RDI}$ exposures have the capacity to regulate the action of proteins catalyzing DNA methylation/demethylation reactions and the $PALB2^{c.1027C > T}$ occurrence, our work is fully in line with the deciphering of the molecular mechanisms by which the exposome is likely to modify the integrity of the genes predisposing to cancer risk (Table 1). Thus, from a mechanistic point of view, our work reinforces the idea that epigenetics can be a source of genetic aberrations promoting cancer. Indeed, several studies show that epigenetic aberrations such as the loss of DNMT1 protein expression or DNMT1/PCNA/UHRF1 complex integrity are the source of genetic instability (deletion, translocation) and point mutations promoting tumor formation [43–45]. The initiator role of environmental factors in epigenetic/genetic-induced tumorigenesis has been already reported. Sciandrello et al., (2004) report that arsenic-induced DNA hypomethylation affects chromosomal instability in mammalian cells [46]. Shimizu et al. (2014) report that the increase of AICDA-catalyzed cytidine deaminase activity in patients with H pylori infection promotes the accumulation of somatic mutations in *TP53* that might promote gastric carcinogenesis [47]. Yoon et al., (2001) report that genetic mutations can occur selectively at methylated CpG sequences in response to polycyclic aromatic hydrocarbons in smoking-associated lung cancers [48].

**Table 1.** Effect of nutraceutical and pollutant mixtures composing our exposome on the presence (or not) of PALB2c.1027C > T mutation.

| Mixtures inducing the $PALB2^{c.1027C>T}$ mutation | Mixtures having a protective effect against the induction of the $PALB2^{c.1027C>T}$ mutation |
|---|---|
| $Folate^{RDI}/Diuron^{3MAC}/PFOA^{3MAC}$ | $Folate^{RDI}/Diuron^{3MAC}/PFOA^{3MAC}/Ascorbic$ $Acid^{3RDI}/Iron^{3RDI}$ |
| $Folate^{RDI}/Zn^{5RDI}/PFOA^{3MAC}$ | $Folate^{RDI}/Zn^{5RDI}/PFOA^{3MAC}/Ascorbic$ $Acid^{3RDI}/Iron^{3RDI}$ |
| $Folate^{RDI}/Diuron^{10MAC}/PFOA^{10MAC}$ $Ascorbic$ $Acid^{3RDI}/Iron^{3RDI}$ | |
| $Folate^{RDI}/Zn^{10RDI}/PFOA^{10MAC}/Ascorbic$ $Acid^{3RDI}/Iron^{3RDI}$ | |

Interestingly, the $PALB2^{c.1027C > T}$ has been described at the germline level in a study on Italian familial breast cancer cases [16]. This mutation was described to be recurrent in the province of Bergamo as it was found in 6/113 (5.3%) familial breast cancer cases and 2/477 (0.4%) controls coming from this area. Germline *PALB2* mutations are inherited in an autosomal dominant manner, of ancestral origin in most cases. To our knowledge, one case of a de novo germline frameshift *PALB2* mutation was described in 2020 [49]. It would be also important to know if the $PALB2^{c.1027C > T}$ has already been described at

an exclusive somatic level as a driver mutation acquired during breast carcinogenesis. Indeed, the identification of environmental factors as potential sources of $PALB2^{c.1027C>T}$ may echo some cases. In addition, the presence of $PALB2^{c.1027C>T}$ in MCF10A cells exposed to Folate$^{RDI}$/Diuron$^{3MAC}$/PFOA$^{3MAC}$ raises the question of the potential cancer driver role of this mutation. Answering this question is very complex and requires the development of a specific research program. However, our data show that subcutaneous injection of MCF10A cells exposed to Folate$^{RDI}$/Diuron$^{3MAC}$/PFOA$^{3MAC}$ does not induce tumor formation as does the injection of MCF10A cells that have lost the integrity of the DNMT1/PCNA/UHRF1 complex [21,45] (Supplementary Figure S1).

Our work also highlights the interest in studying the impact that an environmental factor can have alone but also within cocktails/mixtures. Among the several points illustrating this idea in our work, here is the example of Folate. Indeed, Folate as a single agent is devoid of the ability to promote $PALB2^{c.1027C>T}$, but it appears as a crucial actor of the Folate$^{RDI}$/Diuron$^{3MAC}$/PFOA$^{3MAC}$-induced $PALB2^{c.1027C>T}$ since the absence of this mixture abrogates this mutation presence. Thus, an environmental factor such as Folate can harbor two faces against a cancer-predisposing mutation such as $PALB2^{c.1027C>T}$: neutral in single use or at risk in the Folate$^{RDI}$/Diuron$^{3MAC}$/PFOA$^{3MAC}$ mixtures, as an example. The interest to study the impact of an environmental factor alone is also illustrated in our study by choosing Ascorbic Acid and Iron to redirect DNA methylation/demethylation reactions toward the DNA demethylation pathway catalyzed by TET enzymes at the expense of that catalyzed by APOBECs and TDG. Indeed, without the knowledge of the enhancer role of Ascorbic Acid and Iron in the TET enzyme activity, we would not have investigated the possibility of using these two compounds to counteract the effect of the Folate$^{RDI}$/Diuron$^{3MAC}$/PFOA$^{3MAC}$ and Folate$^{RDI}$/Zn$^{5RDI}$/PFOA$^{3MAC}$ mixture via the induction of the TET enzymes activity. Thus, through all these considerations, our work participates in the debate on the "experimental reproduction" of exposure to mixtures of environmental factors. Indeed, the development of omics analysis techniques allows an increasingly deep characterization of the mixtures of environmental factors to which an individual can be exposed. However, the more these mixtures are composed of entities, the more complex their reproduction in the laboratory is complex since it requires, at least, the consideration of the different "combinations of entities" composing the considered mixture.

Our work also points out the importance of the exposure doses since (i) Folate$^{RDI}$/Diuron$^{MAC}$/PFOA$^{MAC}$ does not induce $PALB2^{c.1027C>T}$ contrary to the Folate$^{RDI}$/Diuron$^{3MAC}$/PFOA$^{3MAC}$ and (ii) Ascorbic Acid$^{3RDI}$/Iron$^{3RDI}$ has a "protective effect" against Folate$^{RDI}$/Diuron$^{3MAC}$/PFOA$^{3MAC}$- and Folate$^{RDI}$/Zinc$^{5MAC}$/PFOA$^{3MAC}$-induced $PALB2^{c.1027C>T}$ but not against Folate$^{RDI}$/Diuron$^{10MAC}$/PFOA$^{10MAC}$- and Folate$^{RDI}$/Zinc$^{10MAC}$/PFOA$^{10MAC}$-induced $PALB2^{c.1027C>T}$ (Table 1). More generally, the dose consideration echoes the question of acute and chronic exposure to environmental factors. In our study, we considered the RDI and MAC as minimum exposure doses, and multiples of these doses were then taken into consideration. The fact of observing an effect with doses equal to three or five times the RDI or MAC may seem high. However, dietary supplements of vitamins, trace elements, ions, or nutraceuticals are likely to contain such doses. Thus, a person following a cure of food supplements could be exposed to doses similar to those used during our study. A similar observation prevails also for the PFOA dose inducing $PALB2^{c.1027C>T}$ since this dose (3MAC: 0.6 ng/mL) is of the same order of magnitude as that detected in the blood of different individuals by several studies (6.78 ng/mL of PFOA are detected in healthy people ($n$ = 194) and PFOA concentration was 9.3 ng/mL with values ranging from 0.8 to 35.2 ng/mL [50,51].

## 4. Conclusions

In conclusion, our work provides a proof of concept supporting the idea that the exposome can generate, by affecting DNA methylation/demethylation reactions, a genetic mutation predisposing to a cancer risk such as PALB2$^{c.1027C>T}$ while discussing the

complexity of conducting such a study in the laboratory through the consideration of the number and dose of compounds selected to mimic exposome.

## 5. Materials and Methods

### 5.1. Cell Culture

MCF10A cells were cultured in DMEM/F12 supplemented with 5% horse serum (Invitrogen, Cergy Pontoise, France), 500 ng/mL hydrocortisone (Sigma-Aldrich, Saint Quentin Fallavier, France), 100 ng/mL cholera toxin (Sigma-Aldrich, Saint Quentin Fallavier, France), 10 µg/mL insulin (ThermoFisher, Courtaboeuf, France) and 20 ng/mL epidermal growth factor (EGF, Sigma-Aldrich, Saint Quentin FallavierFrance), penicillin (100 U/mL), and 2 mmol/L L-glutamine. Glyphosate (CAS 1071-83-6, sc-211568) and Diuron (CAS 330-54-1, sc-239818) were purchased from Santa Cruz (Heildelberg, Germany). DMOG (CAS No: 89464-63-1) was purchased from MedChemExpress (Sollentuna, Sweden). Ascorbic Acid (CAS 50-81-7, A4544), iron (CAS 7782-63-0, F8633), Perfluorooctanoic acid (PFOA, CAS 335-67-1), and Zinc (CAS 7446-20-0, Z0251) were purchased from Sigma (Saint Quentin Fallavier, France). All nutraceuticals or pollutants used in our study are resuspended in PBS/DMSO5% solution.

### 5.2. DNA Extraction

A QIAcube automate and QIAmp DNA Mini QiaCube kit (Qiagen, Courtaboeuf, France) were used to isolate DNA. Qubit (ThermoFisher, Courtaboeuf, France) was used to quantify DNA.

### 5.3. Restriction Site Mutation (RSM) Assay

Digestions were performed with adequate restriction enzymes, HpaII and AciI (NEB, Evry, France). Typically, 1 µg of genomic DNA was digested with 40 U of enzymes at 37 °C for 2 h in 50 µL of reaction. Control samples were treated in the same way but without the addition of the enzyme. Furthermore, 5 µL of digested and undigested mixtures were used for qPCR using QuantiFast SYBR Green PCR Kit and Rotor-Gene Q (Qiagen, Courtaboeuf, France). The primers used in these qPCRs are: S: _cctaaaggtagcagtgaa and as: gcctccaaacttacagg. The mutation level was calculated using Ct values from qPCRs performed with digested and undigested DNA and the $100 \times 2^{-\Delta Ct}$ formula.

### 5.4. Methylated DNA Immunoprecipitation (MeDIP)

After treatment with 1 ng/µL RNase, 5 µg genomic DNA into 130 µL of TE buffer were sonicated with Bioruptor (12 cycles 30 s on/30 s off, Diagenode, Seraing, Belgium). Sonicated DNA was next diluted in 400 µL of TE buffer and denatured in a dry bath heating block for 10 min at 95 °C. On ice, 100 µL of cold 5 × IP and 4 µg of 5 mC antibody (Abcam, Amsterdam, Netherlands) were added to the denatured sonicated DNA. The mixture was then incubated on a rotating platform at 4 °C overnight. 50 µL of magnetic beads (Dynabeads M-280) were washed two times with washing buffer (1 × PBS with 0.1% BSA and 2 mM EDTA) before being resuspended in 50 µL 1x IP buffer. Then, these magnetic beads were added to the antibody-DNA mixture for incubation of 3 h on a rotating platform at 4 °C. After three washes, magnetic beads were resuspended in 250 µL Digestion Buffer and incubated in presence of 3.5 µL Proteinase K (20 mg/mL) for 2–3 h on a rotating platform at 55 °C. DNA purification was next performed by adding 250 µL phenol–chloroform–isoamyl alcohol to each tube. After centrifugation (RT, 14,000× *g*, 5 min), aqueous supernatant was transferred to a fresh microcentrifuge tube in presence of 250 µL of chloroform. After centrifugation (RT, 14,000× *g*, 5 min), aqueous supernatant was transferred to a fresh microcentrifuge tube in presence of 2 µL of the coprecipitant GlycoBlue (20 mg/mL, ThermoFischer, Courtaboeuf, France), 20 µL 5 M NaCl and then 500 µL of 100% ethanol. The mixture was then incubated overnight in a −20 °C freezer. After centrifugation (4 °C, 14,000× *g*, 20 min), the supernatant was removed, and the pellet was washed two times with 1 mL 70% ethanol by incubating at

−20 °C for 10 min then spun again for 10 min. After discarding the supernatant using a pipette and air-drying, the pellet was resuspended in 25 µL of nuclease-free water in order to obtain the MeDIP sample. Then, 2 µL of the MeDIP sample was used for qPCR analysis. qPCR analysis was performed using QuantiFast SYBR Green PCR Kit and Rotor-Gene Q as real-time thermocycler (Qiagen, Courtabouef, France). The primers used in these qPCR are defined in the RSM assay section. Next, the 5-methylcytosine enrichment was calculated using Ct values from qPCRs performed with DNA issue to 5 mC and IgG (a specific antibody) antibody immunoprecipitation and corresponding input sample.

## 6. Elisa

All ELISA kits (5-mC DNA ELISA Kit (Zymo/Ozyme, Saint Cyr l'Ecole, France), Epigenase™5mCHydroxylase TET Activity/Inhibition Assay Kit and Epigenase™Thymine DNA glycosylase Activity/Inhibition Assay Kit (Epigentek/Euromedex, Souffelweyersheim, France), APOBEC3α, APOBEC3γ and TET3 ELISA kit (MyBioSource, San Diego, CA, USA) were used in accordance with the recommendations of the manufacturers.

### 6.1. siRNA Transfection

siRNA-APOBEC3γ (sc-60091, Santa Cruz, Heidelberg, Germany), siRNA-APOBEC3α (sc-72514, Santa Cruz, Heidelberg, Germany), siRNA-TDG (sc-44142, Santa Cruz, France), and siRNA-TET3 (sc-154206, Santa Cruz, Heidelberg, Germany) were transfected in MCF10A as previously described (Duforestel et al. 2019). siRNA-A (sc-37007, Santa Cruz, Heidelberg, Germany), that is, a scrambled sequence devoid of specific degradation of any cellular message was used as control.

### 6.2. Statistical Analysis

All experiments were conducted at least in biological triplicates. Differences in means were assessed using the Student's $t$ test. $t$. $p < 0.05$ was considered significant (*).

**Supplementary Materials:** The following supporting information can be downloaded at: https://www.mdpi.com/article/10.3390/epigenomes6040032/s1, Figure S1: Tumorigenicity of MCF10A cells exposed to Folate$^{RDI}$/Diuron$^{3MAC}$/PFOA$^{3MAC}$.

**Author Contributions:** P.-F.C. designed experiments and coordinated the project; F.C., G.B.-C., and P.-F.C. performed all experiments. All authors (F.C., G.B.-C., C.A., J.-S.F. and P.-F.C.) interpreted and discussed the data. P.-F.C. wrote the manuscript. All authors have read and agreed to the published version of the manuscript.

**Funding:** This work was supported by grants (EpiSAVMEN—DynaS-PDL2016) from REGION PAYS DE LA LOIRE.

**Institutional Review Board Statement:** Not applicable.

**Informed Consent Statement:** Not applicable.

**Acknowledgments:** F.C. was supported by fellowships from the "EpiSAVMEN/REGION PAYS DE LA LOIRE" research program. This paper was prepared in the context of the SIRIC ILIAD program supported by the French National Cancer Institute National (INCa), the Ministry of Health and the Institute for Health and Medical Research (Inserm) (SIRIC ILIAD, INCa-DGOS Inserm-12558).

**Conflicts of Interest:** The authors declare that they have no competing interest.

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
