# Peer review of "Modulation of DNA Methylation/Demethylation Reactions Induced by Nutraceuticals and Pollutants of Exposome Can Promote a C > T Mutation in the Breast Cancer Predisposing Gene PALB2"

_2075-4655_

Round 1
Reviewer 1 Report
In this work, Courant et al tested the effects of different compounds on the regulation of DNA methylation/demethylation mechanisms that may ultimately lead to C>T mutations, taking as example a mutation in PALB2 gene. As the authors claim, their results provide a proof of concept about the role of exposome in the dysregulation of the DNA methylation machinery with a possible induction of cancer driver mutations.
I have some comments that need to be addressed.
- In the title, I would suggest avoiding the redundancy of “induce”.
- In the abstract, please specify that “TET-mediated conversion of 5-methylcytosine” consists in DNA de-methylation.
- In the introduction, some concepts are unnecessarily explained at the expense of others that are crucial to understand the present work. For example, the concept of cancer driver mutation is well known, but if the authors want to remind this concept, “stop codon” needs to be omit in the sentence. On the other hand, very few information is given about DNA methylation/demethylation reactions and their regulation, as well as detail about the compounds chosen to be tested. As far as I understand, PALB2 gene has been selected as an example of cancer predisposing gene. I would suggest to make this clear in the manuscript since the introduction in order to reinforce the authors’ message to provide a proof of concept that may be translated to other cancer driver mutations. Moreover, the literature about the role PALB2 mutations and studies investigating PALB2 methylation in breast cancer need to be cited in the introduction.
Results:
- Statistical significance must be reported both in the text and the figures.
- Please always specify “folate supplementation” both in the text and in the legend of figure 3.
- In the first paragraph of the results, regarding the sentence “As expected, the quantification of 5-methylcytsoine (5mC) by ELISA method” please specify that global DNA methylation level was investigated.
- Please explain the criteria adopted to focus the analysis on the four C>T mutations mentioned.
- Please report the use of methylated DNA as a control shown in Figure 3.
- In figure 6, I suggest to indicate (with a box or an arrow) the combination of compounds giving the results described.
Materials and methods:
- other cell lines other than MCF10A cells are described in the paragraph of cell cultures but no information has been given about their use in the experiments.
Finally, I believe that it would be interesting to investigate the effects of PALB2 mutation as possible driver of cancer transformation and I suggest to mention this aspect in the discussion even if the authors cannot perform these functional experiments.
Minor comments:
- Figure 2 is cited only in the paragraph related to folate supplementation but describes the entire experimental plan.
- Sometimes, English too informal and needs to be revised using more formal language. Please carefully check and correct errors in the text (e.g., “exemple” in the introduction, 5-methylcytsoine in the results, m5C in the legend of figure 3)
- Acronyms of should be defined the first time they are introduced. Please also define the acronyms for molecular biology techniques, compounds, genes, proteins the first time they are named in the text.
- Please follow the Guidelines for Human Gene Nomenclature (now some gene names are not italicized).
- Please check reference citation style.
- Please check the use of “previous” in the description of MeDIP method.
Author Response
Point-by-point response to the reviewer’s comments - In the title, I would suggest avoiding the redundancy of “induce”. >> Based on this remark, we have corrected the title of our manuscript. - In the abstract, please specify that “TET-mediated conversion of 5-methylcytosine” consists in DNA de-methylation. >>We thank the reviewer for this comment. TET-mediated conversion of 5-methylcytosine is implicated in the early steps of a more global DNA de-methylation process. However, TET-mediated conversion of 5-methylcytosine is not strictly speaking a DNA demethylation. To avoid any confusion, we would not prefer not to modify the abstract. - In the introduction, some concepts are unnecessarily explained at the expense of others that are crucial to understand the present work. For example, the concept of cancer driver mutation is well known, but if the authors want to remind this concept, “stop codon” needs to be omit in the sentence. >> We thank the reviewers for this comment. We have now modified our introduction. On the other hand, very few information is given about DNA methylation/demethylation reactions and their regulation, as well as detail about the compounds chosen to be tested. >> Based on this remark, we have added a paragraph to introduce the DNA methylation/demethylation. As far as I understand, PALB2 gene has been selected as an example of cancer predisposing gene. I would suggest to make this clear in the manuscript since the introduction in order to reinforce the authors’ message to provide a proof of concept that may be translated to other cancer driver mutations. Moreover, the literature about the role PALB2 mutations and studies investigating PALB2 methylation in breast cancer need to be cited in the introduction. >> In agreement with these two comments, we completed our introduction by writing "In our work, the choice of the PALB2 gene as a "demonstrator gene" is based on the fact that this gene is methylated or mutated in different types of cancer including breast cancer. Thus, the results obtained for this "demonstrator gene" may be transposed to other genes. Results: - Statistical significance must be reported both in the text and the figures. >> Based on this remark (which echoes those of other reviewers), we have indicated the statistically significant differences mentioned in our manuscript. - Please always specify “folate supplementation” both in the text and in the legend of figure 3. >> Thanks to the reviewer for pointing out this point of inaccuracy in our article. The use of the word "supplementation" revising the use of the different folate’s RDI used, we have corrected our article accordingly. - In the first paragraph of the results, regarding the sentence “As expected, the quantification of 5-methylcytsoine (5mC) by ELISA method” please specify that global DNA methylation level was investigated. >> Thanks to the reviewer for pointing out this point of inaccuracy in our article that we have corrected on the revised version of our manuscript. - Please explain the criteria adopted to focus the analysis on the four C>T mutations mentioned. >> Based on the remark, we have added on the revised version of our manuscript that the choice of these 4 mutations is based on the fact that these mutations must be included in a “methylable” CG dinucleotide. - Please report the use of methylated DNA as a control shown in Figure 3. >> We thank the reviewer for drawing our attention to this lack of precision in the legend of Figure 3. We have remedied this by mentioning that "Methylated DNA" refers to the use of Universal Methylated DNA Standard (Ozyme/Zymo, France). - In figure 6, I suggest to indicate (with a box or an arrow) the combination of compounds giving the results described. >> We thank the reviewer for this proposal to improve the reading of figure 6. For this purpose, we have added bars to guide the reader to the main results mentioned. Materials and methods: - other cell lines other than MCF10A cells are described in the paragraph of cell cultures but no information has been given about their use in the experiments. >> Thank you for pointing out this error that we have corrected in the revised version of our manuscript. Finally, I believe that it would be interesting to investigate the effects of PALB2 mutation as possible driver of cancer transformation and I suggest to mention this aspect in the discussion even if the authors cannot perform these functional experiments. >> We thank the reveiwer for this interesting and timely question. Indeed, the investigation of the role of PALB2 driver mutation is totally relevant. In anticipation, and because we are strongly interested in the environmental causes of cancers (our previous work showed that exposure to diuron and glyphosate could be a source of tumors in two-hit oncogenic models Briand et al. 2019 and Duforestel et al. 2019), an experiment was performed in the laboratory in which MCF10A cells exposed to FolateRDI/Diuron3MAC/PFOA3MAC were injected subcutaneously in nude mice. As shown in Supplementary Figure 1, no tumors were detected in mice injected with these cells. This result therefore suggests that exposure of MCF10A cells to FolateRDI/Diuron3MAC/PFOA3MAC is not tumorogenic on its own, although this exposure is capable of promoting the PALB2c.1027C>T mutation. However, the role of mutation driver of PALB2c.1027C>T cannot be fully determined by this result. For all these reasons, we have written the following paragraph in the discussion of our article " the presence of PALB2c.1027C>T in MCF10A cells exposed to FolateRDI/Diuron3MAC/PFOA3MAC raises the question of the potential cancer driver role of this mutation. Answering this question is very complex and requires the development of a specific research program. However, our data show that subcutaneous injection of MCF10A cells exposed to FolateRDI/Diuron3MAC/PFOA3MAC does not induce tumor formation as does injection of MCF10A cells that have lost the integrity of the DNMT1/PCNA/UHRF1 complex ". Minor comments: - Figure 2 is cited only in the paragraph related to folate supplementation but describes the entire experimental plan. >> We thank the reviewer for pointing out this lack. We have therefore referred to Figure 2 for all other treatments. - Sometimes, English too informal and needs to be revised using more formal language. Please carefully check and correct errors in the text (e.g., “exemple” in the introduction, 5-methylcytsoine in the results, m5C in the legend of figure 3) >> >> Thanks to the reviewer for pointing out these errors that we have corrected in the revised version of our manuscript. - Acronyms of should be defined the first time they are introduced. Please also define the acronyms for molecular biology techniques, compounds, genes, proteins the first time they are named in the text. - Please follow the Guidelines for Human Gene Nomenclature (now some gene names are not italicized). - Please check reference citation style. >> All these three suggested corrections were performed in the revised version of our manuscript. - Please check the use of “previous” in the description of MeDIP method. >> Based on this point, we have reformulated the description of MeDIP method.Reviewer 2 Report
0.For the convenience of review, please add line numbers in the manuscript. To add line numbers in Word, first click the “Layout” tab in the Ribbon. Then click the “Line Numbers” drop-down button in the “Page Setup” button group to display a drop-down menu of line number options. Please select the "Continuous" option when submitting the manuscript.
When pointing out the Manuscript, it is very difficult without line numbers.
1.If you think Exposome is a usually used keyword in scientific fields, why not suggest “exposome” in the title?
2.It would be good to add that seven exposomes were used in Abstract.
“Compounds of our exposome” → “Compounds of our seven exposome”
3.Abstract has an unusual italic sentence. From “Ascorbic acid and … to “in the PALB2 gene”
4. The Introduction is well-written in a way that is appropriate in length and content, and is intended to capture the reader's interest. However, the very last sentence of the second paragraph is strange. Also, the last citation in the third paragraph is not in format.
5.There is not enough explanation for the legend in Figure 2. What do the arrows mean? Additionally, citations for MAC and RDI are required. And do you have MAC and RDI for zinc?
6.Pleas modify the layout of Figure 3 →
Not
Figure 3A
Figure 3B
But
Figure 3A Figure 3B
It should be arranged in the latter. There are too many blanks in their current state. And in Figure 3B, are there statistical processing results for each group? Can you represent this in the form of an asterisk? Method P < 0.05 was marked as an asterisk, but none of the figures in this paper were marked.
7.Figure 4 should also show the difference by group through statistical processing. MAC-dependent changes are interesting, but there must be a statistical basis to explain them.
8.In Figure 6, in order to present the MAC and RDI dependent changes, statistical processing is a prerequisite. I did not find any statistical basis for a concentration-dependent change.
9.The content and volume of the Discussion is appropriate. However, if some of the contents of the second and fifth paragraphs are arranged in a table, readers can check them more intuitively.
10.Overall: genes and restriction enzymes should be marked as italic.
Author Response
Point-by-point response to the reviewer’s comments 1.If you think Exposome is a usually used keyword in scientific fields, why not suggest “exposome” in the title, >> We thank the reviewer for introducing this point. Although arguable, the use of the word "exposome" seems appropriate for the title of our paper. Thus on the basis of this point, we propose the following title for the revised version of our manuscript: Modulation of DNA methylation/demethylation reactions induced by nutraceuticals and pollutants of exposome can promote a C>T mutation in the PALB2 breast cancer predisposition gene. 2.It would be good to add that seven exposomes were used in Abstract. “Compounds of our exposome” → “Compounds of our seven exposome”. >>Thank you for this proposal that we have followed in the revised version of our maniscript. 3.Abstract has an unusual italic sentence. From “Ascorbic acid and … to “in the PALB2 gene” >>Thanks to the reviewer for pointing out this error which we have corrected in the revised version of our manuscript. 4. The Introduction is well-written in a way that is appropriate in length and content, and is intended to capture the reader's interest. However, the very last sentence of the second paragraph is strange. Also, the last citation in the third paragraph is not in format. >> We thank the reviewer to underline this mistake. We have rephrased this last sentence by writing « [13]. In addition to be detected in breast cancer, literature reports that the presence of these mutation in healthy people were associated with a high-risk of breast cancer”. We have also corrected the format of the situation in question 5.There is not enough explanation for the legend in Figure 2. What do the arrows mean? Additionally, citations for MAC and RDI are required. And do you have MAC and RDI for zinc >> Thanks to the reviewer for drawing our attention to the lack of precision in the legend of figure#2. To remedy this, we have completed our legend by writing: « The cells were exposed to six doses of the compounds over three weeks (2 doses per week during 3 weeks) ». We have also added references to support MAC and RDI (please see new references section). 6.Please modify the layout of Figure 3 → Not Figure 3A Figure 3B But Figure 3A Figure 3B It should be arranged in the latter. There are too many blanks in their current state. And in Figure 3B, are there statistical processing results for each group? Can you represent this in the form of an asterisk? Method P < 0.05 was marked as an asterisk, but none of the figures in this paper were marked. >> Based on these comments, we have redesigned Figure 3 and added the statistics supporting our results. We have also added new sentences in the Figure3’s legend to guide readers in interpreting this figure. 7.Figure 4 should also show the difference by group through statistical processing. MAC-dependent changes are interesting, but there must be a statistical basis to explain them. >> Thanks for this remark. We have added new sentences in the Figure3’s legend to guide readers in interpreting this figure. 8.In Figure 6, in order to present the MAC and RDI dependent changes, statistical processing is a prerequisite. I did not find any statistical basis for a concentration-dependent change. >> Thanks for this important comment. We have added statistic and new sentences in the Figure6’s legend to guide readers in interpreting this figure. 9.The content and volume of the Discussion is appropriate. However, if some of the contents of the second and fifth paragraphs are arranged in a table, readers can check them more intuitively. >> Based on this comment, we have added a table (Table 1: XXX) to summarize the effect of our exposome mixtures on the presence (or not) of PALB2c.1027C>T mutation (please see Table 1). 10.Overall: genes and restriction enzymes should be marked as italic. >>This point has been corrected in the new version of our manuscript.Reviewer 3 Report
The paper describes the effect of folate, diuron, zinc, PFOA, ascorbic acid, iron, glyphosate on the methylation/demethylation and mutation of a selected sequence related to driver mutations of PALB2 gene using MCF10A cells as an in vitro model. This hypothesis-driven study uses adequate methods and conclusions are supported by results. Altogether, this is a very interesting study showing that the exposome may lead to cancer-related mutations by affecting DNA methylation profile.
There are several minor remarks that need to be addressed:
Figure 1 - the conversion of 5-hmC to C does not occur only in a passive way (TET).
Figure 2 - just to be clear, the cells were exposed to six doses of the compounds over three weeks?
Figure 3 & 4 & 5 & 6 - were any of the observed changes statistically significant? If so, this is not shown in the graph.
For additional comments, see the attached file.

Author Response
Point-by-point response to the reviewer’s comments Figure 1 - the conversion of 5-hmC to C does not occur only in a passive way (TET). >> Thanks for this constructive comment. We have indicated on figure 1 the existence of an active 5hmC>C conversion without however detailing it here. Figure 2 - just to be clear, the cells were exposed to six doses of the compounds over three weeks? >> Thanks for this remark echoing remark #5 of review#1. To answer it, we have completed our legend by writing: « The cells were exposed to six doses of the compounds over three weeks (2 doses per week during 3 weeks). Figure 3 & 4 & 5 & 6 - were any of the observed changes statistically significant? If so, this is not shown in the graph. >> Based on these comments, we have added statistics to strengthen our data and their interpretation (please see new version of our figures). For additional comments, see the attached file. >> The spelling and grammatical errors reported in the pdf document have been corrected >> We have completed the references of the NCI’s dictionary of cancer terms by adding https://www.cancer.gov/publications/dictionaries/cancer-terms. >> The reference of article published by Breast Cancer Association Consortium was corrected. >> DNMTs, TETs, TDG and HDACs have been defined. >> The decitabine concentration used in our experiments was indicated (10μM). >> Figure 6A was corrected.Round 2
Reviewer 2 Report
I confirmed all figures, and the authors well revised my opinions.
In my opinion, the manuscript is acceptable.